# A Static Hybrid Renewable Energy System for Off-Grid Supply

**Augusto Montisci** [1,*] and **Marco Caredda** [2]

1   Department of Electrical and Electronic Engineering, University of Cagliari, 09123 Cagliari, Italy
2   Independent Researcher, 09048 Sinnai, Italy; marco.caredda@gmail.com
*   Correspondence: augusto.montisci@unica.it

**Abstract:** The electrification of the rural areas of the planet has become one of the greatest challenges for sustainability. In fact, it would be the key to guaranteeing development for the poorest areas of the planet from which most of the raw material for the food market derives. The paradigm of centralized production is not applicable in these territories, because the distribution network would involve unjustifiable costs. For this reason, many studies have been carried out to ensure that the energy supply (specifically electricity) for off-grid utilities is maintained, in order to guarantee energy autonomy while reducing dependence on specialist assistance for the management of the system. In this work, a hybrid system (HRES) is proposed that combines the exploitation of solar radiation, wind power, and biomass using static devices, in order to improve the system's availability and limit the cost of operation and maintenance. The aim of the study is to define promising lines of research, which can improve the sustainability of renewable harvesting systems to supply off-grids users.

**Keywords:** HRES; energy static conversion; off-grid supply; thermoacoustics





## 1. Introduction

The problem of sustainability has recently taken on prominence on the agenda of almost all national governments and international institutions, both because of the increasingly evident human impact on the ecosystem, with increasingly dramatic consequences, and the push of the public who are becoming increasingly aware of the risks involved. Unfortunately, the policies adopted so far have limited themselves to allocating financial resources to programs of a very general nature, and there is still a long way to go to significantly affect the footprint. In fact, what has been achieved so far is only to reduce the relationship between the footprint and economic growth. However, the challenge that humanity must face today is to reverse the trend, while guaranteeing resources for all the populations that inhabit the whole planet. A simple reduction in consumption would be feasible, but not likely. In fact, the states that consume the most are also those with the greatest political weight, and they continue to adopt policies that place sustainability in the background of economic growth. While waiting for the states to take note of the fact that the economy cannot grow indefinitely, a change in the growth paradigm itself can be made immediately, shifting the focus on quality rather than quantity. This is possible because awareness of the inadequacy of economic indicators is growing more and more, and real well-being is often a countertrend with respect to economic wealth. Energy is likely the sector in which it is easier to raise awareness among the population, and, at the same time, the one in which results can be achieved more easily. In fact, at present, there is a very wide gap between the ways in which energy is used and the available technology. Reducing this gap would result in an immediate reduction in the consumption of resources and in the impact that their use entails, but for a period that can be estimated to be a few decades, this would produce, rather than decrease, economic growth. It has been amply demonstrated that the replacement of fossil fuels with renewables involves an increase in the number of people employed, and even if the turnover of companies is reduced, the economic balance benefits from it. Furthermore, the growth in employment feeds the market, which does not

lead to a worsening of the footprint, as long as the entire economy is sustainable. The use of renewables is recognized as one of the fundamental factors of implementing a sustainable economy, but it is not the only one. The real keyword is "smartness," or the reasonableness in identifying technical solutions. For example, today, the centralized energy model, in which a power plant produces energy and then extensively distributes it to a network of users, on a national and often supranational scale, is increasingly in crisis. The limitations of this model are mainly three: the primary source is normally a fossil fuel, which is not extracted near the power plant, and, therefore, it is necessary to spend energy on transport; the power plant requires the exclusive use of a much wider area than the plant, having to ensure that the distance from the external area is appropriate; third, the transmission and distribution network is an extremely expensive infrastructure, and it is estimated that 20% of the total electricity produced is lost in distribution. It is therefore understandable that there is a rising interest in solutions such as smart grids, where the three problems described, rather than being solved, are eliminated at the root.

The study presented in this work is part of this trend, but paying particular attention to the specific problem of the electrification of off-grid areas [1–8]. Electricity is considered to be a fundamental driver of development; however, in many rural areas of the planet, the creation of a distribution network would be completely unjustified. The literature includes numerous studies concerning the supply of off-grid utilities or microgrids using renewable sources. Not surprisingly, these studies mostly focus on the rural areas of Africa and India, and on areas with many islands, such as the Philippines [9–14]. The problems that must be faced are the low density and discontinuity of the primary source, and the cost of energy storage systems [15–18]. There are many tools that allow one to cope with these difficulties. The main approach is making consumption smart; that is, to make use of efficient devices, to consume energy for real needs, and to adapt the timing of consumption to limit the need to store energy.

Having said this, the energy system must maximize its sustainability, both in environmental and economic terms [19–25]. The sun is by far the most widely used primary source, to produce both heat and electricity. In domestic installations and small smart grids, photovoltaics is used almost exclusively [26–30], while thermodynamic solar is commonly used in power plants. The choice of smart solutions ensures that thermal users are powered by converting solar radiation directly into heat, rather than powering an electric heater that is then powered by electricity. At the same time, it would be reasonable to carry out cooling by directly exploiting the heat, but this type of solution is less common. Another primary source widely used in residential areas is wind [31–33]. The market offers various solutions for vertical axis micro-wind turbines, installed on the ground. Other solutions, such as geothermal energy, sea energy, and rivers, are applicable only in certain cases, but, on a global scale, they can offer an important contribution. Perhaps the most problematic aspect is energy storage. The options are several (batteries, ultracapacitors, flywheels, counterweights, heat tanks, etc.), but each has limitations, meaning that optimal load management is extremely important. To mitigate the storage problem, several hybrid solutions have been proposed in the literature, in which different primary sources and different storage systems are incorporated into a single system (Hybrid Renewable Energy Systems; HRES [34–39]). In Figure 1, a general flowchart of an HRES system is represented. The optimal layout depends on many factors, such as the available primary sources, energy demand, and cost constraints, meaning it is not possible to define the design criteria of general validity. The most immediate advantage of HRES derives from the fact that the different sources are not correlated with each other in time; therefore, in a multi-source system, the intervals during which there is no production are shorter, meaning the necessary capacity can be reduced.

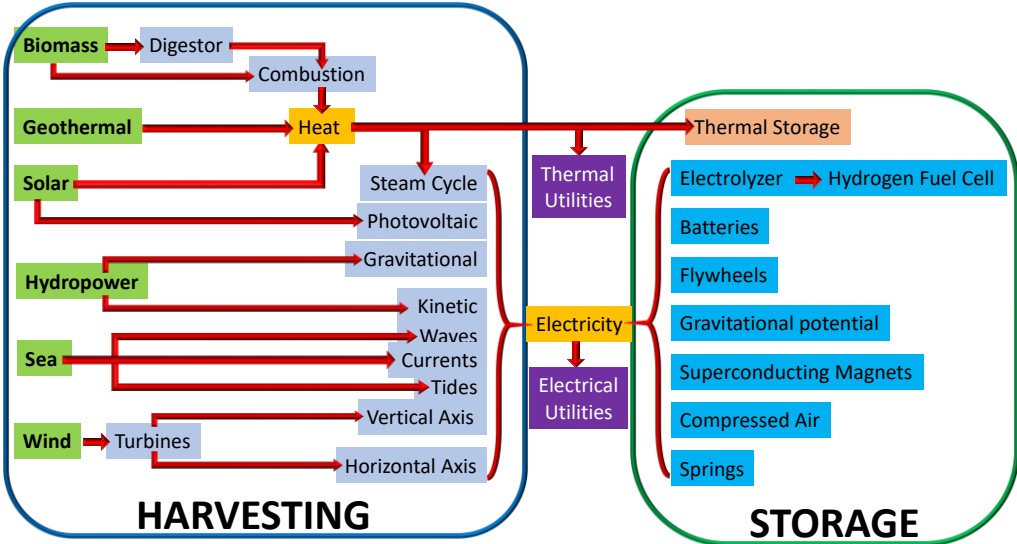

**Figure 1.** General scheme of HRES system.

Since the target areas of the studies on renewables mainly concern the poorest regions of the earth, the economic aspect plays a fundamental role in the definition of technical solutions [40–48]. For this reason, studies almost always foresee an optimization problem, which can normally be formulated in linear terms. The HOMER software [49], developed by the U.S. National Renewable Energy Laboratory (NREL), represents a standard for the optimization of HRES systems. The optimal solution depends on the specific context in which the system is implemented; therefore, the costs of the various technologies, as well as the availability of sources, vary from one region to another.

One of the objectives of the present study is to identify technological solutions that allow the costs of the installation and management of the system to be reduced, thus increasing their economic feasibility. It is, therefore, not a question of seeking an optimal combination of devices available on the market, but, rather, defining alternative technologies for individual processes, in such a way as to reduce the specific costs of the individual parts of the system.

The common feature of the proposed solutions is that of providing exclusively static devices (i.e., without moving parts). This property generally makes it possible to create systems with high performance, long life, low maintenance costs, and high energy density, both with respect to volume and mass. The main reason for this is the absence of any dry friction, while other reasons include the ability to operate at very high temperatures, the lack of inertia, and the absence of constraints on the shape of the system. If there are more options to statically convert a primary source, a choice is still made based on efficiency, cost, decay of performance, and decommissioning at the end of the lifecycle. This study focuses on the primary sources available in rural areas (in which the study is oriented), including sun, wind, and biomass, with the last essentially consisting of crop residues and forest foliage.

The rest of the article is organized as follows. Section 2 describes the processes and devices used for both conversion and storage. In Section 3, the layout as a whole is presented, and a broad sizing of the system is indicated. In Section 4, the assessments of the study are motivated in light of a comparison with a layout that adopts the most common technical solutions. Some final remarks conclude the article.

## 2. Materials and Methods

The guiding idea of this study is to use static devices for the conversion of energy deriving from different types of primary sources, as well as for their storage and recovery. Static and quasi-static energy conversion processes have always attracted the attention of

both the research world and industry, due to their considerable advantages: the absence of friction and wear, longer life-cycle duration, reduced maintenance, the possibility to operate in a contaminated environment and to use sources at extremely high temperatures, higher power density (both with respect to volume and mass), and greater operational flexibility. Nonetheless, only a limited number of these processes have been adopted on a large scale, as each of them have different problems and there is already an alternative technology that has had a strong development. For this reason, the comparison between traditional and innovative static mobile technologies must be made with these considerations in mind, based on the potential of the future evolution that can be hypothesized for each. Photovoltaics deserves a separate discussion. It is, in fact, a static technology that converts solar radiation into electricity in a single stage; thus, it would seem to meet all the requirements for this study. However, static nature is the only requirement that this technology possesses among those required here, as it lacks in terms of sustainability. In fact, the cost per kWh is very high, the efficiency decays in a few years, the maintenance is expensive, it is not suitable for operating in different environments, it presents decommissioning problems, and, finally, the balance between the energy produced in the life cycle and that consumed to build a panel is roughly even. For this reason, in this work, an alternative technology is adopted for the conversion of solar energy, which, as will be seen below, has further important advantages.

This section will analyse, in detail, the components of the general layout. Some of the technologies described below have already reached a very advanced level of development; other components are still at a low TRL, but have, nevertheless, been included in this work as they possess better characteristics of sustainability compared to traditional competitors, both from a technical and an economic point of view. For example, this is the case of the thermoacoustic generator, which is preferred to photovoltaics for the exploitation of solar energy by virtue of the lower cost, or the use of resonant cavities instead of turbines for the exploitation of wind energy. This choice, although it may seem like a gamble in the view of an investment, offers interesting prospects and, for this reason, it is believed to be worth analysing. The prospect is to obtain a highly integrated and completely static hybrid system, which, overall, should guarantee the system a considerably higher technical–economic sustainability standard than that of the HRES systems proposed in the literature.

### 2.1. Solar Concentrators

Solar energy is used here for three different types of utilities; namely, heat generation, refrigeration, and electricity generation. Regardless of the utility to be powered by solar radiation, it is always advisable to concentrate it in advance [50,51]. Ideally, this solution could also be used for photovoltaics, because, for the same amount of captured energy, the active surface would be reduced. Nonetheless, in practice, it is not preferable to adopt this type of solution, because photovoltaic cells pose serious cooling problems (which would be accentuated in this way), despite there being several studies and experiments in this direction that often combine the production of electricity with heat. This problem does not exist when the entire spectrum of solar radiation contributes to the transformation process, as in the cases examined in this work. Through non-imaging concentrators, it is possible to concentrate the solar radiation to reach high temperatures, with the benefit of efficiency. Different types of concentrators exist, although, in this work, the Fresnel lenses and mirrors are used, which, compared to parabolic and hyperbolic concentrators, have the advantage of a lower encumbrance. The main difference between mirrors and lenses is that, in the latter, the radiation must pass through a medium, which can never be completely transparent. On the other hand, with mirrors, the focal point is visible, resulting in a visual impact and possible maintenance problems. The receiver, positioned in the focus of the concentrator, must be able to withstand very high temperatures without rapidly degrading, and, at the same time, must limit the energy radiated as a result of the temperature. If the temperature of the receiver remains low, one can think of exploiting the greenhouse effect—encapsulating the receiver inside a vacuum tube in a material that

is transparent to solar radiation, but opaque with respect to the radiation emitted by the receiver itself. As the temperature rises, the overlap between the solar spectrum and the emission spectrum of the receiver becomes greater, meaning a compromise must be found between the energy lost by the radiation of the receiver and the solar energy that does not reach the receiver because it is filtered through glass. In this case, it is preferred to insert the receiver inside a cavity, so that the ratio between emitted energy and incident energy depends on the geometry of the cavity. The in-depth study of this element goes beyond the scope of this study; thus, one will hypothesize that the system can reach a temperature of 500 °C, referring to the specific literature for the design aspects.

### 2.2. Heat Loads

The water temperature for domestic utilities is much lower than that which can be reached even with a simple solar thermal panel. Conversely, in the case of electricity or refrigeration production, there is an interest in maximizing the temperature at the receiver. For this reason, it makes sense to combine the different utilities powered by the sun, to ensure that the exhaust heat coming from the two high temperature processes feeds the users that require hot water (domestic water, kitchen, washing machine, dishwasher, etc. (40 °C–100 °C)). For the sake of simplicity, a single heat accumulation system at a constant temperature (110 °C) is assumed, where the thermo vector fluid must avoid creating hazards to human health.

### 2.3. Solar Fed Electrical Generation

As an alternative to photovoltaics, the thermoacoustic process (TA) is preferred in this work, by virtue of lower costs and greater efficiency. The thermoacoustic effect [52–56] was discovered by the Phoenician glass blowers as early as the first century BC; however, it is only recently that the phenomenon has been interpreted. Experimentally, they observed that, when the difference of temperature at the ends of the duct overcame a critical value, a sound was produced. The phenomenon is quite simple: the hot wall transfers some heat to a particle of gas which, as a consequence, enlarges its volume, making it touch a colder part of the wall, and then return the heat to the wall, reducing its volume and assuming the initial position. The oscillation of the particles is perceived by the human ear as a sound, and it represents a conversion of heat into mechanical energy (gas vibration), meaning it performs as an engine without solid moving parts. In order for the phenomenon to last over time, it is necessary to ensure that the two ends of the duct are kept at a constant temperature, by exchanging heat with two sources at a high and low temperature, respectively. The low temperature heat exchange can take place with air, or with a fluid of exchange. Since the efficiency of the thermoacoustic conversion depends on the temperature gradient, and not on the absolute value, an element (termed a regenerator) is usually inserted inside the resonant tube. The regenerator is made of a low thermal conducting material, and has a shape that ensures the gas particles can swing freely in the longitudinal direction of the tube. The adoption of the regenerator allows the thermoacoustic effect to trigger, even with modest temperature jumps (100 °C). It is understood that the greater the overall temperature gradient, the higher the conversion efficiency will be. Another fundamental aspect concerns the gas that is used as an operating fluid. Since the gas is contained within the resonator, it is possible to choose a gas with properties that are better suited to the process. Typically, the gases used for this type of device are helium and argon.

After the first thermoacoustics stage, in order to obtain an electrical current, a further transformation is necessary. This can be, again, performed statically, by means of piezoelectric crystals, or by transferring the TA vibration to a solid component. The former solution has the advantage that the entire conversion process is static, but the level of power is limited. On the other hand, by interfacing the TA stage with an alternate cycle, such as a Stirling engine, the level of power is higher, but the static nature of the whole apparatus is compromised. An easy solution for this stage consists of using a loudspeaker in reverse

mode, in the sense that the vibration moving the magnet of the cone induces an electrical current in the coil.

### 2.4. Solar Cooling

The thermoacoustic effect is reversible in the sense that, by means of a sound vibration, it is possible to pump the heat, similar to a cooling cycle (see Figure 2). This means that, by coupling a thermoacoustic resonator with a thermoacoustic cooler, the energy of the sun can be used for cooling. This solution is much more preferable to that of producing electrical energy to power a conventional appliance, because many energy conversion steps are avoided, alongside the benefit of the efficiency and the cost. Furthermore, the simplicity of the thermoacoustic device is exploited to perform the entire transformation from the source (sun) to the utility.

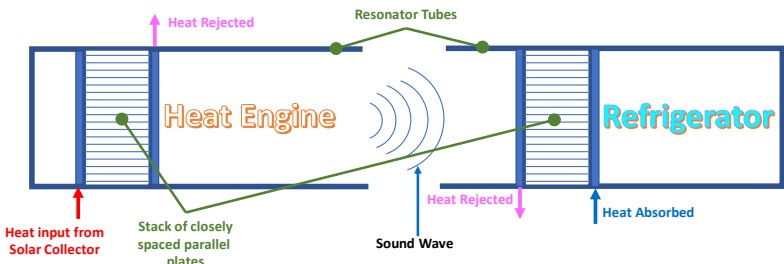

**Figure 2.** Thermoacoustics-based cooling system.

### 2.5. Wind Power

The possibility of tapping into different primary sources of energy is of fundamental importance for an off-grid system. In fact, any renewable source remains unavailable for shorter or longer periods, during which it is necessary to make up for it through reserves previously accumulated or through conventional generation systems (typically the generator set). In both cases, the periods of unavailability of the primary source generally represent the greatest obstacle to overcome for the realization of a totally autonomous system.

Typically, wind energy is preferred in extra urban areas, due to the better ratio between the produced energy and land used, as opposed to other renewable energy sources. Therefore, the integration of wind energy conversion systems into buildings is a small but growing trend, and it has great scope in generating electricity from the wind. In fact, in Europe, there is increasing interest from home owners and businesses to install small wind turbines on the rooftops of houses, school buildings, hospitals, commercial buildings, or even on tall buildings [33]. Small wind turbines are particularly suitable for the distributed production of electrical energy. Combining solar and wind systems allows one to mitigate the problem of the discontinuity of the primary source, which is typical for both sources alone. On the other hand, it is important to improve the ratio between the encumbrance and the power of the system. In [32], a system is proposed that exploits the building to convey the wind towards a small turbine placed in the under roof. This innovative system allows one to avoid the typical drawbacks of wind generation, such as the encumbrance of the blades, noise, cut-in and cut-off velocities of the wind, turbulence, etc. Such layout entails several advantages. Firstly, the visual impact and safety problems of the turbine are definitively avoided, because it is hidden inside the structure. Secondly, the wind direction is not a problem, because the vertical axis of the rotor allows it to capture the wind regardless of which direction it comes from. Thirdly, the stator allows one to handle a wide section of flow; it is possible the entire building can be used to intercept the wind, even if the dimension of the turbine is very small, depending on the ratio between the inlet and the outlet cross section of the stator. This allows one to exploit a wider range of wind velocities as opposed to common wind generators, due to a lower cut-in and unlimited cut-off velocities.

The power associated with the wind flow is given by the formula:

$$P = \frac{1}{2} \rho \, A \, v^3 \tag{1}$$

where $P$ represents the power, $\rho$ is the air density, $A$ is the cross section of the flow tube, and $v$ is the wind speed. From (1), it emerges that the time diagram of wind speed significantly affects the power associated with the wind. More specifically, it can be seen that, with the same average speed, the power available is greater the more the speed values are variable. For this reason, the site where a wind power plant is installed is preferably characterized with the Weibull diagram [31], which provides the probability density function of the wind speed as a function of two parameters that characterize the site. It follows that the possibility of processing the wind, regardless of its speed, determines a significant variation in the electrical power produced, specifically in case the statistical distribution of the speeds has a significant tail beyond the cut-off speed. The system proposed in [32], due to its small size and the fact that it is constrained into a cage, allows the turbine to rotate at much higher regimes than any ordinary wind turbine. Only a part of the kinetic power of the wind can ideally be converted by a wind turbine, because the flow out of the turbine cannot be stopped, otherwise the conversion would cease. Betz's law establishes that the maximum amount of power is obtained when the output speed is equal to 1/3 of the input speed. In the case described in [32], Betz theory is not immediately applicable, because a centripetal turbine is adopted, in which the conversion of the wind power is not linked to the reduction of speed, but to the change of direction from horizontal to vertical. For this type of system, it is not possible to formulate a theory of general validity, as the properties of the incoming and outgoing flow depend on the boundary conditions, and, therefore—regarding the case in question—on the shape of the building and the orography of the territory. Nonetheless, the different way of processing the flow compared to conventional wind turbines favors the conversion process, meaning it can be assumed that the limit set by Betz's law can be exceeded.

In this work, in order to create an entirely static system, the turbine is replaced by resonant cavities [57]. In this way, wind energy is converted into acoustic power, in exactly the same way as for solar energy, meaning it is possible to integrate the two sources to create a single transformation system, both for electricity and for cooling. The evaluation of conversion efficiency in this case is even more complex, due to the high number of degrees of freedom in the sizing of the resonant cavities. The sizing of the system goes beyond the aims of this study, which only aims to provide an order of magnitude of the system parameters.

### 2.6. Energy Storage

An energy storage system is necessary because the system is off-grid. As mentioned above, the need to store energy is mitigated by the fact that two different primary sources are harvested, which reduces the time intervals in which no external source is available. The need for a smaller reserve is reflected in the lower cost of the storage system. The static and hybrid nature of the system is also adopted for energy storage. The variety of storage systems allows each device to be sized in order to minimise both costs and space. For this reason, in principle, it would be convenient to use non-static devices, such as flywheels and counterweights. However, for consistency with the static nature of the system, these solutions are not taken into consideration in this work. Batteries are also excluded, due to their limited lifespan and the disposal problems they entail. For these reasons, the storage system considered will be a combination of a cycle based on hydrogen, ultracapacitors, and biomass. The individual modules are briefly described individually below.

### 2.6.1. Hydrogen Energy Storage

Although the hydrogen-based storage system has a high cost compared to all the other components of the plant, it has the advantage that there is no limitation on the

amount of energy it can store, other than that of the hydrogen containers. Excess electricity supplies an electrolyser that decomposes water into oxygen and hydrogen. To draw from the energy reserve, the hydrogen is oxidized by means of a fuel cell, obtaining a direct current, which can eventually be transformed into power by the inverter. For both the electrolyser and the fuel cell, the market offers a wide range of solutions of different sizes, efficiency and, therefore, price. It is important that, in the dimensioning of the hydrogen storage system, the real need is not exceeded, while also taking into account the other storage systems in order to avoid an unnecessary increase in costs. The hydrogen system also has the advantage of not entailing the release of greenhouse gases into the atmosphere. A lower cost solution, although also less efficient, is the one that provides the combustion of hydrogen, feeding the same thermoacoustic system that converts the solar energy. This solution allows investment costs to be significantly reduced, because the fuel cell is replaced with a simple burner. However, while the former can reach an efficiency of up to 70%, the efficiency of the thermoacoustic cycle is approximately 10%. This implies that, in order to guarantee a sufficient reserve of energy, it will be necessary to increase the power of the system, as well as provide a greater volume of hydrogen storage. Furthermore, it is not recommended to use air as an oxidizer for hydrogen, because this creates nitrogen oxides, which are among the main causes of acid rain. Since, ideally, the stoichiometric amount of oxygen necessary for the combustion of hydrogen is obtained as a by-product of hydrolysis, it is sufficient to store this oxygen and use it for combustion. However, this requires a greater amount of oxygen storage, and the burner is also more complex because the combustion temperature is significantly higher.

### 2.6.2. Ultracapacitors

Ultracapacitors (UC) [58] represent an alternative to batteries, as they store electrical energy through an electric field, rather than through a chemical reaction. Although the power density of these devices is not competitive with respect to batteries, they have very significant advantages. This includes the possibility of carrying out charge/discharge cycles in a very short time, without any consequence on the duration of the life cycle, which is significantly longer than batteries. The market offers numerous alternatives to ultracapacitors, which can be installed and used in the same way as a battery. In combination with the hydrogen system mentioned in the previous paragraph, the UCs allow for the better management of current transients, both in the hydrogen production phase and in the use of reserves. If used for this purpose, the required capacity becomes modest, as does the cost. Alternatively, it is possible to think about the operating principle of the UCs to build a system capable of storing much greater quantities of energy. The energy stored in a capacitor is given by the formula:

$$E = \frac{1}{2}CV^2 \tag{2}$$

where $E$ is the energy, $C$ the capacity, and $V$ the potential difference between the armatures of the capacitor. The technology of UCs, compared to that of traditional capacitors, has made it possible to increase the energy that can be stored by some orders of magnitude. This result was obtained following modifications in the realization, which allowed the capacity $C$ to be increased. This is expressed by the relation:

$$C = \varepsilon\frac{A}{d} \tag{3}$$

where $\varepsilon$ is the permittivity of the insulating material placed between the plates, $A$ is the area of the two plates, and $d$ the distance that separates them. The increase in capacity was obtained by adjusting the two parameters of the area and the distance. To increase the former, the capacitor plates were made of porous material, meaning that the active surface can be up to 3000 times the value of the external surface. On the other hand, to reduce the distance $d$, an electrolyte with a suitable concentration of ions was inserted in place of

the dielectric between the plates. In this way, when the voltage *V* is applied between the plates, the ions of the electrolyte are attracted to the armature with opposite polarity, thus obtaining a double layer capacitor (i.e., two capacitors in series where the two intermediate plates are constituted from the electrolyte ions). To allow the ions to get as close as possible to the two plates, they were not electrically isolated, meaning that the distance *d* separating the plates is reduced almost to an atomic scale. To avoid the short circuit between the plates and ions, the operating voltage of the capacitor must be small— typically 6 V. To obtain higher voltage values, the capacitors were connected in series.

In this work, it was attempted to exploit the operating principle of the UCs to create a system that is not currently available on the market and does not present particular implementation difficulties. The idea is to use ultra-capacitive panels to cover all the internal surfaces of the house, such as the walls, the ceiling, and the floor. While using a less sophisticated technology than the UCs currently on the market are made with (and, therefore, limiting costs), it is possible to create a storage system with a capacity that is far superior to that of any normal battery system. Keeping in mind the principle of sustainability, which advocates not to make plants larger than is required, and by virtue of the fact that the use of different primary sources reduces the need to store energy, it can be assumed that such a system can, alone, provide the storage capacity needed by a domestic user. Nonetheless, to increase the resilience of the entire system, it is considered appropriate to promote the adoption of different storage technologies.

2.6.3. Biomass

Specifically in poorer countries, biomass (mainly firewood) is still the most used primary source of energy [59]. In various regions of the world, this has a negative impact on the environment, specifically in areas where a process of desirtification is underway. For this reason, attempts are made to favor the transition to other forms of energy, sometimes even preferring fossil fuels. Taking this aspect into account, biomass has not been included among the main sources of the proposed system, although they are considered as reserve. It is worth distinguishing between wet biomass and dry biomass. The use of the former does not cause the impact on the environment mentioned above, and, indeed, its use for energy purposes mitigates the problem of waste disposal. The simplest way to use wet biomass for energy purposes is to use an anaerobic digester, which produces a biogas that can be burned to produce the heat with which to feed the thermoacoustic cycle, or to power thermal utilities (sanitary water, kitchen, washing). In the same way, dry biomass can also be used. In this work, to avoid overly complicating the layout of the system, it was decided to only use dry biomass, therefore removing the need to set up an anaerobic digester.

**3. Layout of the Overall System**

This section presents a working hypothesis of a static HRES system for meeting the energy needs of an isolated housing unit. While respecting the constraints set out above, regarding the exclusive use of static devices, and the attention to sustainability, both from a technical–economic and environmental point of view, the definition of the system has wide margins of discretion. Furthermore, the design solution strictly depends on the climate of the area in which the project is carried out, the specific energy demand of the user, and which materials and professional skills are most easily accessible. Having said this, it is clear that any layout presented here can only represent an initial draft, which would require several changes to adapt it to any real context.

Figure 3 shows a functional diagram of the integrated system. The roof cover is made up of Fresnel lenses that heat a coil through which the thermoacoustic generator is powered. The heat taken from the cold end has a sufficient temperature to power the building's thermal utilities (sanitary water, heating, washing). It is possible to switch the thermoacoustic resonator to choose whether to produce electricity through the loudspeaker or to power a second thermoacoustic resonator for cooling (food preservation, conditioning). The thermoacoustic system can also be powered by a stove, which works as a backup on

the main reserve represented by hydrogen. The volume below the roof is divided into converging ducts to convey the wind towards the resonant cavity in the central column. The walls of the building are shaped in such a way to converge the flow of wind towards the roof and, at the same time, limit turbulence at the inlet to the duct. In the diagram in Figure 3, for simplicity, only one of the four ducts is reported. The wind resonant cavity is interfaced with the thermoacoustic resonator, in order to share the next stage of conversion of the acoustic power.

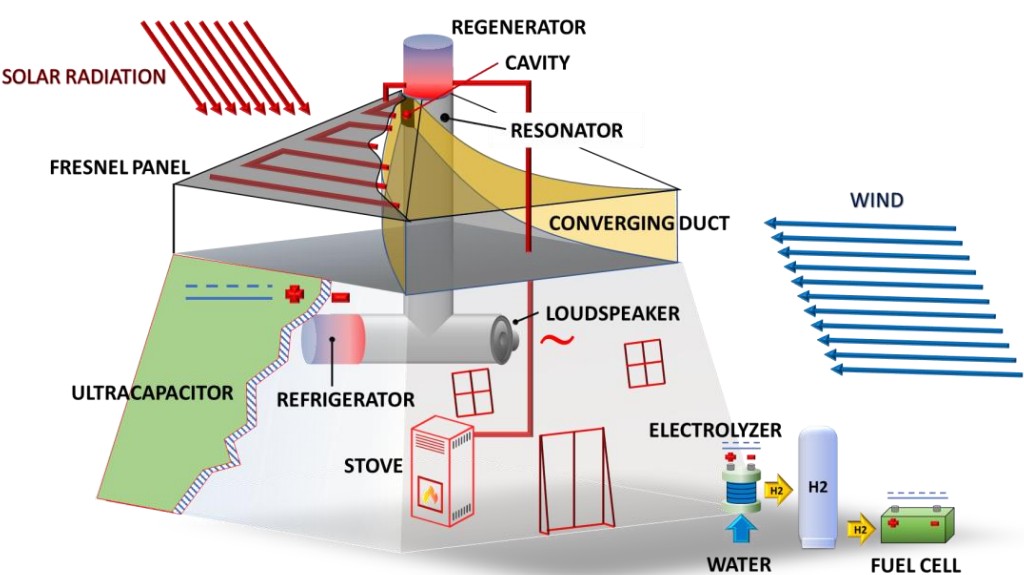

**Figure 3.** Functional scheme of the proposed HRES system.

When the electricity produced exceeds consumption, it can be stored both by means of the ultra-capacitor panels and by powering an electrolyzer for the production of hydrogen. Ultracapacitors are suitable for storing small amounts of energy and, in the presence of strong transients, in both the charging and discharging phases. Conversely, hydrogen allows the storage of a much greater amount of energy, but requires more regular operation. The hydrogen produced by the electrolyser is stored in a special container under pressure and is used to power the fuel cell when consumption exceeds production.

*System Dimensioning*

It is difficult designing a system that meets the requirements without oversizing it. First of all, it is necessary to consider the hourly diagram of loads in the different periods of the year, also taking into account the hourly flexibility of the loads, which allows the amount of stored energy to be limited. Preliminarily, it is assumed that the energy demand is equal to 36 MJ/day (divided between electrical, thermal, and refrigeration loads), without further specification on the hourly diagram, even if it is reasonably assumed that consumption is more concentrated during daylight hours.

Subsequently, it is necessary to evaluate the producibility of energy from the sun and wind. Solar energy has the advantage of regularity, which favors sizing, but has the disadvantage of being available for a limited period within the 24 h. Depending on the geographical area, solar radiation can be more or less relevant. In this study, it was decided to build the plant on the island of Sardinia, in the Mediterranean, where the average value of daily solar radiation is approximately 25 MJ/m$^2$. It is assumed that the Fresnel panels completely cover the roof of the building, and that, during the hours of sunshine, the surface exposed to solar radiation is equal to 10 m$^2$, for a total of 250 MJ/day. On the basis of the literature data, an efficiency of 90% of the concentration, 30% of the thermoacoustic conversion, and, finally, 37% of the loudspeaker is assumed, with an overall efficiency equal to 10%, for which 25 MJ/day is obtained to supply the electrical utilities. Part of

this energy is absorbed for refrigeration, for which an overall efficiency similar to that of electricity generation is assumed as a first approximation. An amount of 90% of the thermal energy processed by the thermoacoustic generator is recovered on the cold side of the regenerator, at a temperature of approximately 100 °C, thus being suitable for powering thermal utilities. By assuming a lower temperature on the cold side of the regenerator, it is possible to slightly increase the efficiency of the thermoacoustic stage. Therefore, by modulating the supply temperature of the thermal utilities, it is possible to vary the amount of electricity produced.

To correctly size the wind system, as mentioned above, it is necessary to consider numerous elements, such as the fluid dynamic characteristics of the building, the orography of the surrounding area, and the wind regime. As a first approximation, it is assumed that the building is able to convey a flow tube of 20 m$^2$ into the converging duct. The overall wind energy depends on the distribution of speeds, while the wind direction does not affect the process. On the basis of in situ monitoring data, it was possible to define the following parameters of the Weibull distribution of the wind at zero altitude: $k = 1.7$, $\lambda = 4.2$. Assuming the Betz formula, a mean wind energy value of 114 MJ/day is obtained. In this work, one cautiously assumes that the conversion efficiency of kinetic energy into vibration is 50% and, considering that the loudspeaker is the same as is used in the thermoacoustic system (37% efficiency), an additional electrical energy value of 21 MJ/day is obtained by the wind system. This means that, on average, the electricity generated would be theoretically sufficient to supply all the loads. This is not the reality, as the surplus of energy over the average consumption is necessary to cope with the losses in storage.

The smart organization of loads takes on fundamental importance because, as long as it is possible, it is better not to use storage. The loads must be divided between those that can be deferred and those that cannot be, and it must be ensured that the latter are active in the hours in which there is a greater production of electricity. This means that the loads must be scheduled mainly during the day, limiting the night loads as much as possible. At the same time, it is useful to have a certain number of loads that can be activated at any time the wind is blowing; therefore, possibly even at night.

The part of production that exceeds the consumption is directed to the storage system. This is done in three different ways; namely, the ultracapacitor panels, the integrated electrolyser + fuel cell system, and biomass. None of the three systems have the characteristics to be adopted as the only solution, although their complementarity allows them to be combined effectively.

The ultra-capacitive panels have very low losses in the charge/discharge cycle and can withstand very high currents, which makes them suitable for managing current peaks, in both the charging and discharging phases. The commercial ultracapacitors can reach energy densities of up to 200 MJ/dm$^3$, although this value is not plausible for capacitive panels such as those used in this study. Considering non-engineered systems made in the laboratory, an energy density of 1 MJ/dm$^3$ can be considered plausible. To realize a paneling with the dimensions of 100 × 20 × 0.1 dm$^3$, one obtains a system capable of storing approximately 200 MJ, which, for the working hypotheses, corresponds to approximately 5.5 days of autonomy. This value cannot be considered sufficient, as the maximum consumption occurs precisely in the winter period, when the availability of solar energy is reduced, while the energy of the wind is not very different from that which occurs in the summer; therefore, the exhaustion of this reserve is a plausible hypothesis. Theoretically, the number of panels could be increased, but the solution appears to be excessively expensive. More accurate economic data will be obtainable when panels such as those described in this paper will be available on the market.

The part of the energy to be stored that exceeds the capacity of the panels is accumulated in the form of hydrogen by means of an electrolyser. For the Electrolyser + Fuel Cell system, there are various solutions on the market, even for domestic solutions. In this work, it was decided to use a 2.2 kW electrolyser, capable of producing up to 0.5 m$^3$/h of hydrogen, which corresponds to 6.22 MJ, which, in turn, can be converted into 3.24 MJ by

means of the fuel cell. The amount of energy that can be stored depends on the hydrogen storage volume. This represents the long term energy reserve, and serves to compensate for the imbalance between production and consumption, which typically occurs during the winter. Therefore, the sizing of the storage volume must be calibrated considering the possibility of having to accumulate energy for the winter. It should also be considered that the overall efficiency of the complete charge/discharge cycle is approximately 41%, while in the case of the panels, the efficiency is close to unity. Therefore, the use of this form of storage must be evaluated carefully and only used to the extent request.

The third mode for energy storage involves biomass. With this system, it is possible to power the thermoacoustic resonator in place of the sun and, in this way, the electric utilities are powered. The system is considered in this work for emergency use, in the event that the other two reserves are exhausted, but it does not fall within the energy balance of the system, which is sized in such a way that the two aforementioned systems are sufficient to guarantee to meet the demand of energy.

## 4. Discussion

In light of what has been described in the previous section, it is appropriate to analyze the components of the proposed system one by one, to justify their use as an alternative to the solutions normally adopted in the literature (Figure 4).

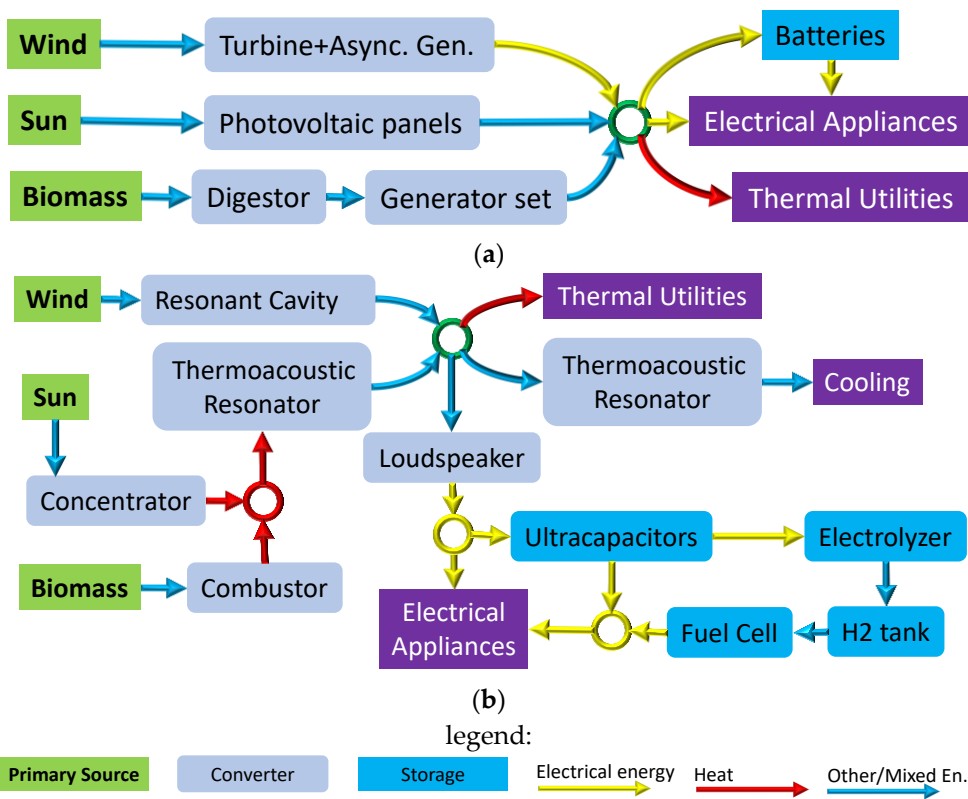

**Figure 4.** Conventional (**a**) and proposed (**b**) scheme of the proposed HRES system.

As for the wind system, the proposed technical solution has the twofold advantage of using the entire building to capture the wind, and replacing the turbine with a static conversion system. The lower conversion efficiency, compared to the turbine, is therefore compensated for by elaborating a flux tube with a much larger section than that of a commercially available vertical axis turbine. It would, theoretically, be possible to adopt the solution proposed in [32], which involves the use of a small vertical axis turbine in the center of the roof. Compared to this solution, a static solution has been preferred in this study, for the reasons explained above related to maintenance, availability, and life cycle—all of which are elements of extreme importance in the case of off-grid users. On the

other hand, the possibility of using a large amount of primary energy makes the need to maximize conversion efficiency less pressing.

As for the conversion of solar energy, it is important to justify a choice other than photovoltaics. This, in fact, has all the static requirements set out in this study, and allows the co-generation of electricity and heat to be carried out. In reality, photovoltaics presents some problems relating to sustainability, such as cost, the decline in efficiency in a few years, and issues in decomissioning. Furthermore, the production of photovoltaic cells requires a large amount of energy, approximately equal to what a cell produces over its entire life cycle. Taking into account that the production processes are mostly powered by fossil fuels, it is understood that, even in global terms, the PV is not able to significantly affect the consumption of resources. Another important aspect concerns the useful share of the solar spectrum, which is very limited in the PV, while in the thermoacoustic system it is used entirely. Therefore, with the same active surface, the quantity of primary energy processed is greater. The cost is, certainly, an element in favor of the thermoacoustic system, as the surface of the panels has the sole task of concentrating the solar radiation. Therefore, it can be made with materials with low technological content, whether it is reflective surfaces or translucent materials. The option of using solar concentrators for PV systems is rarely adopted, because it further complicates the already difficult problem of heat dissipation, for which almost all of the projects require the entire collection surface to be made up of photovoltaic cells. Finally, in the thermoacoustic solution, the electricity is generated in alternating forms, which allows adjustments to be made to the voltage level for subsequent uses, or for storage, without involving the inverter.

In order to minimize maintenance costs and the need for external technical assistance, in this study, it has been assumed that biomass is used directly as a fuel. Therefore, the use of the anaerobic digester is not envisaged, unless it is often provided for off-grid installations coupled with a generator set that produces both heat and electricity. The biomass taken into consideration is limited to residual crops and foliage (generally available in rural areas), while the exploitation of organic waste for energy production is not foreseen.

An important feature of the proposed system is that the three primary sources are all converted into vibration energy, before being transformed into electrical energy by a single loudspeaker. The fact that the resonator replaces both the wind turbine and the generator set powered by biogas, and the louspeaker replaces the two corresponding alternators, compensates for the greater complexity of the solar radiation conversion chain compared to the PV. Heat cogeneration is inherent in the thermoacoustic transformation, powered by both solar radiation and biomass, and the temperature level depends on the regulation of the cooler end of the regenerator. Utilities such as air conditioning and cooling can be powered directly (i.e., without the conversion into electricity followed by the power supply of a special appliance [56]), with benefit in terms of efficiency. As for storage, the binary system consisting of the ultra-capacitive panels and the hydrogen-based system, is adopted here as an alternative to the more common batteries. Batteries are simpler to use than the hydrogen system, and meet the static conversion requirement, but have limits on the duration and energy that can be stored, and have end-of-life disposal problems. The system consisting of the electrolyser and the fuel cell is often adopted in off-grid systems, mainly due to the fact that the capacity is fixed by the storage volume of tanks, and not by the size of the electrolyser. However, it should be considered that the performance on the charge/discharge cycle is much lower than that of batteries, and it is not suitable for sudden changes in current intensity. For this reason, storage using ultracapacitors was envisaged in this study, which have a charge/discharge efficiency equal to the batteries, and can withstand very high instantaneous currents. Rather than using commercial products, this study envisages the creation of ultra-capacitive panels that can be applied to the walls of the building and compensate for the lower energy density with a greater volume. The aim is still to limit the costs of the system to increase its economic sustainability. From a functional point of view, if the costs were suitable, nothing changes if the panels are replaced with commercial products.

## 5. Conclusions

In this work, a hybrid solution is developed for an off-grid system powered by renewable energy sources. The purpose of the study is to combine technologies that maximize the sustainability of the system, both from an environmental and an economic point of view. In this regard, various aspects are analyzed, such as the cost of installation and maintenance, the energy balance over the entire life cycle, and the decay of performance over time. The potential of emerging technologies is taken into consideration, rather than technological maturity, in the sense that some solutions that require further development in some cases are preferred to others that have a consolidated position on the market. The common feature of the proposed technologies is the absence of moving components, in order to avoid dry friction and wear. Another important aspect is related to the cost of materials, which leads to the search for alternative solutions, aside from photovoltaics, for electricity generation and batteries for storage. The sources used are solar radiation, biomass, and wind; all of which are converted into acoustic power through a thermoacoustic resonator in the first two cases, and a resonant cavity in the third. The sound energy is, in turn, transformed into electrical energy through a loudspeaker, or for cooling through a second thermoacoustic resonator that is coupled with the first one in reversible mode. The excess energy produced is stored through a dual system consisting of ultra-capacitive panels, which are used for short-term storage, and a combined electrolyser + fuel cell system, which guarantees the reserve in the inter-seasonal period. The study is still in a preliminary phase and requires experimental verification. However, it offers promising prospects for the electrification of isolated users.

**Author Contributions:** Data curation, A.M. and M.C.; Formal analysis, A.M.; Investigation, A.M. and M.C.; Methodology, A.M. and M.C.; Resources, M.C.; Writing—original draft, A.M.; Writing—review & editing, A.M. and M.C. All authors have read and agreed to the published version of the manuscript.

**Funding:** This research received no external funding.

**Institutional Review Board Statement:** Not applicable.

**Informed Consent Statement:** Not applicable.

**Data Availability Statement:** Data sharing not applicable.

**Conflicts of Interest:** The authors declare no conflict of interest.

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
