# Peer review of "A Static Hybrid Renewable Energy System for Off-Grid Supply"

_sustainability, doi:10.3390/su13179744_

Round 1

Reviewer 1 Report

Dear Authors,

Thank You for the opportunity of reading this article.

General statements about the article:

-> The article discusses the HRES that combines the exploitation of solar energy with that of the wind through the use of static devices, in order to improve the system's availability and limit the cost of operation and maintenance. Thus the topic and scope of the article are interesting, actual, and highly desirable.

-> The article content suite to Sustainability journal scope.

-> abstract is adequate to article content

-> Keywords are correctly proposed.

-> Organization of the paper is clear and correct.

-> Literature review is based on 60. They are related to article content. A sufficient number of them are actual.

However, I indicated the following elements to revision:

#1

The results are well presented. But there is a luck of discussing them. Thus I recommend adding a separate section with discussion.

#2

For Section 2, Please introduce the proposed research framework more effective, i.e., some essential brief explanation vis-à-vis the text with a total research flowchart or framework diagram for each proposed algorithm to indicate how these employed models are working to receive the experimental results. It will increase the value of the presentation.

#3

Please extend conclusions. I suggest to add a paragraph with limitations of the proposed approach as well as future research directions in conclusions section.

#4

Please also revise the manuscript regarding the personal way of addressing the text. Please avoid and replace we" or "our" with the impersonal manner of addressing. The text will sound much more professional.

Technical issues:

-> Please add required additional information like: author contribution,  data availability statement, etc.

Author Response

The authors wish to thank the Reviewers for their suggestions, which allowed to significantly improve the quality of the paper.

In the following the answers to the remarks are reported

Reviewer 2 Report

In this manuscript, authors proposed a hybrid renewable energy system (HRES) for off-grid supply. Even though the readers has enough interested in the proposed system, the significant faults are included for publishing as follows:

  1. Contrary to huge literature survey in the manuscript, we cannot find any clues on why to propose this kind of HRES. Please reduce unnecessary related work and compare the proposed system with previous work directly.
  2. Contribution is significant weak because the explanation for the proposed system is not enough. Please modify Abstract and contribution of Introduction. Specially, the existing problems to provide the proposed solution were broadly mentioned. Please be summarized more detailedly.
  3. The subsection for 2.6.3. Biomass is required to be removed due to being out of scope in this manuscript.
  4. Please be kindly mentioned for abbreviation.
  5. Numbering of some equations are omitted.

Author Response

(The authors gave the same response as above.)

Round 2

Reviewer 1 Report

Dear Authors,

Thank You for the revision.

Ally my prepositions were included.

Thus I recommend publishing this article in present form.

Best regards,

Reviewer

Reviewer 2 Report

Authors have improved the revised manuscript for reviewer's comments. 

Thus, I'd like to accept the publication of the revised manuscript.

However, before the publication I suggest authors kindly add Abbreviations in front of References for Sustainability readers.